# Tendon and Ligament Injuries in Elite Rugby: The Potential Genetic Influence

**DOI:** 10.3390/sports7060138

**Published:** 2019-06-04

**Authors:** Jon Brazier, Mark Antrobus, Georgina K. Stebbings, Stephen H. Day, Shane M. Heffernan, Matthew J. Cross, Alun G. Williams

**Affiliations:** 1Sports Genomics Laboratory, Department of Sport and Exercise Sciences, Manchester Metropolitan University, Manchester M1 5GD, UK; mark.antrobus@northampton.ac.uk (M.A.); g.stebbings@mmu.ac.uk (G.K.S.); a.g.williams@mmu.ac.uk (A.G.W.); 2Department of Psychology and Sports Sciences, University of Hertfordshire, Hatfield AL10 9AB, UK; 3Sport, Exercise and Life Sciences, University of Northampton, Northampton NN1 5PH, UK; 4Faculty of Science and Engineering, University of Wolverhampton, Wolverhampton WV1 1LY, UK; stephen.day@wlv.ac.uk; 5Physiotherapy and Sports Science, University College Dublin, D04 V1W8 Dublin, Ireland; shane.heffernan@ucd.ie; 6Premiership Rugby, Twickenham TW1 3QS, UK; mcross@premiershiprugby.com; 7Institute of Sport, Exercise and Health, University College London, London WC1E 6BT, UK

**Keywords:** genomics, rugby, polymorphisms, soft-tissue injury, tendinopathy, ligament rupture

## Abstract

This article reviews tendon and ligament injury incidence and severity within elite rugby union and rugby league. Furthermore, it discusses the biological makeup of tendons and ligaments and how genetic variation may influence this and predisposition to injury. Elite rugby has one of the highest reported injury incidences of any professional sport. This is likely due to a combination of well-established injury surveillance systems and the characteristics of the game, whereby high-impact body contact frequently occurs, in addition to the high intensity, multispeed and multidirectional nature of play. Some of the most severe of all these injuries are tendon and ligament/joint (non-bone), and therefore, potentially the most debilitating to a player and playing squad across a season or World Cup competition. The aetiology of these injuries is highly multi-factorial, with a growing body of evidence suggesting that some of the inter-individual variability in injury susceptibility may be due to genetic variation. However, little effort has been devoted to the study of genetic injury traits within rugby athletes. Due to a growing understanding of the molecular characteristics underpinning the aetiology of injury, investigating genetic variation within elite rugby is a viable and worthy proposition. Therefore, we propose several single nucleotide polymorphisms within candidate genes of interest; *COL1A1, COL3A1, COL5A1, MIR608, MMP3, TIMP2, VEGFA, NID1* and *COLGALT1* warrant further study within elite rugby and other invasion sports.

## 1. Introduction

Due to the characteristics of the game of rugby, whereby high-impact body contact frequently occurs through multiple physical collisions and tackles, musculoskeletal injuries are extremely common [1,2]. Rugby Union (RU) has one of the highest reported incidences of match injuries within professional sports, regardless of the injury definition used [3]. This is likely in part due to the well-established and frequently applied injury surveillance research compared to other collision sports. Rugby League (RL) does not currently have a comparable level of injury surveillance research which limits our understanding somewhat. The majority of injuries in both RU and RL occur during tackles [1,4,5,6,7,8]. However, numerous other causes have been documented, including but not limited to rucks, mauls, scrums [9] and via tripping, twisting, slipping, falling, overexertion, and overuse [10]. A meta-analysis by Williams et al. [11] reported the total incidence of injury (injuries per 1000 player h) as 81/1000 in matches (~3 injuries per match) and 3/1000 in training in men’s professional RU. 

The regular occurrence of injury in RU limits competitive success. For example, Williams et al.’s [12] recent seven-year prospective study assessing playing time loss from injury and team success in elite RU found clear negative associations between injury measures (injury burden and injury days per match) and team success (league points tally and Euro Rugby club ranking). Thus, reductions in injury incidence and severity could enhance team success. 

Due to the high incidence of injury in RU, numerous injury surveillance studies have been conducted during international competitions, particularly during the five Rugby World Cups from 1995 to 2015 [7,8,13,14], as well as single and multiple seasons for professional [6,15,16] and community level rugby [17,18]. Although numerous injury surveillance studies have been carried out in RU, only studies from 2007 were consistent with the international consensus statement for epidemiological studies in rugby [19]. Therefore, comparisons with earlier studies are problematic. This consistency has not existed to the same degree for RL, although recent steps have been taken towards a consensus-driven approach [20].

Injury data collection is an essential part of trying to understand the risk (incidence and severity) of participation in sports and how that risk changes over time. van Mechelen et al. [21] designed a four-step model for injury prevention within sport. It involves (i) identifying the extent of the sports injury problem, (ii) identifying the characteristics and mechanisms that contribute to the development of injury, (iii) introducing measures to reduce future risk and/or severity of injury, (iv) an evaluation of those measures by repeating the first step. A similar risk management model was proposed by Fuller and Drawer [22], which aimed to identify risk factors and estimates that could be evaluated and then communicated to the sports community. Having a deeper understanding of these areas enables coaches, doctors and strength and conditioning staff to assess current practices in injury prevention, treatment, and rehabilitation, and make adjustments accordingly. It also allows governing bodies to identify areas of high risk and to introduce strategies to mitigate such a risk. Finally, longitudinal injury data allows researchers to monitor the impact and effectiveness of any interventions. What is apparent from the research undertaken thus far is that injuries vary considerably in location, diagnosis and profile.

Fundamental understanding of injury mechanisms and differences in inter-individual risk begins with the genome and the biological composition of tissues that depend on coordinated expression of selected genes at the protein level. Being able to utilize these genetic data alongside the traditional injury prevention practices may enable a more personalized approach to injury risk management at the elite level of rugby. The aim of this narrative review, therefore, is firstly to highlight the incidence and severity rates of tendon and ligament injury within elite rugby. Secondly, to discuss the biological composition of tendons and ligaments and how genomics may influence this and subsequent predisposition to injury. The steps necessary to better understand the genomic aspects of injury within elite rugby will then be considered. A structured literature search was performed for empirical research studies and review articles. The search terms included “elite rugby injury”, “injury mechanisms in rugby”, “tendon and ligament epidemiology”, “pathophysiology of tendon and ligament injury”, “molecular characteristics of tendon and ligament injury”, “genetics of tendon and ligament injury”. The reference lists of all articles were also examined for eligible studies to minimize the possibility of relevant articles being omitted.

## 2. Tendon and Ligament Injury Incidence Rates and Severity in Rugby

### 2.1. Tendon and Ligament Injury Incidence Rates in Rugby

Numerous injury surveillance studies have been carried out within professional RL, with muscle/tendon and ligament/joint (non-bone) injuries consistently the two most frequent types of injury [4,23,24,25,26,27]. However, the majority of professional RL studies are dated, have limited application to present day RL, and inconsistent methodological approaches and definitions were used. Cross et al. [28] demonstrated the importance of utilizing consistent definitions for injury by showing that incidence of injury with a >24-hour time-loss definition was approximately double that when using a >7 day definition. For example, Gissane et al.’s [23] injury definition was “the onset of pain or a disability resulting from either training for or playing rugby league,” while Seward et al.’s [24] definition was “that which caused a player to be unavailable for selection in a match, or participation in a training session or any other injury which required medical treatment, other than routine conservative measures.” These differences provide substantially different portrayals of injury risk. When the injury definition is more exclusive and includes only more severe injuries, joint/ligament injuries are most frequent. However, when the definition is more inclusive, muscular, head and neck injuries are most frequent [2]. This has led to much debate on definitions of injury within RL [29,30,31]. A very recent attempt was made at a consensus-driven approach to standardize epidemiological studies in RL [20], and these data are probably more valid than those previously reported. Three different ligament injuries were in the top five for incidence: medial collateral ligament (MCL) 3.9/1000 h, syndesmosis 2.7/1000 h, ankle lateral ligament 2.6/1000 h [20].

In RU, injury incidence rates are easier to identify than RL due to the consensus statement on injury definitions and data collection procedures for studies in RU [19]. However, much like RL, muscle/tendon and ligament/joint (non-bone) injuries are consistently the top two most frequently occurring injury groups in elite RU [6,7,8,11,14,32] with more muscle/tendon injuries in backs than forwards at English Premiership and International level. For ligament/joint (non-bone) injuries, forwards appear to have more frequent occurrence at international level, while backs have more at English Premiership level [6,7,14]. It should be noted, however, that these apparent differences between forwards and backs are based on data provided in the literature but not statistical testing. Table 1 summarizes the match injury incidence of muscle/tendon and ligament/joint (non-bone) injuries from post-2007 studies where methodologies align with the consensus statement on injury definitions and data collection procedures [19]. It is worth noting that, at World Cup competitions, although muscle/tendon injuries have a high incidence, this is mainly due to the presence of muscle rather than tendon injuries [7,8,14]. It is likely that this also occurs in the English Premiership and Super 14 competitions, but the data are not clear.

In the English Premiership RU competition across the seven most recently reported seasons from 2011–2018, ligament injuries were consistently amongst the top five most common injuries [16,37,38], with MCL in the top five every season apart from 2015–2016. The Professional Rugby Injury Surveillance Project (PRISP) reports individual injuries such as MCL, hamstring or ankle lateral ligament, rather than grouping all muscle/tendon or ligament/joint (non-bone) injuries together. Outside of the top five injuries, there are no available data on further muscle/tendon and ligament/joint (non-bone) injuries, making more detailed or grouped analysis impossible. Figure 1 shows the top five most common match injuries in the English Premiership competition during 2011–2018, highlighting the frequency of ligament injuries.

### 2.2. Tendon and Ligament Injury Severity and Burden in Rugby

The current literature is limited regarding the severity (days absence from full training or match play) of injuries at specific anatomical locations in elite RL. From the available data, Gibbs [40] found ankle ligament tears were the most severe, followed by MCL tears and groin muscle/tendon tears. More recently, Orchard [41] stated anterior cruciate ligament (ACL) tears were the most severe, followed by shoulder sprains and dislocations and MCL tears. This is supported by Fitzpatrick et al. [20], although that study calculated severity from date of occurrence until date of return to full training, which differs from the RU’s consensus statement on injury definitions and data collection procedures [19] and would increase severity data. These studies suggest that ligament/joint (non-bone) and muscle/tendon injuries are the main causes of RL players missing matches, thus impairing competitive success and player wellbeing.

Rugby union has similar but more consistent findings to RL, with muscle/tendon and ligament/joint (non-bone) injuries making up three of the top five most severe injuries for forwards; ACL, Achilles tendon and MCL injuries caused 988, 726 and 718 days absence, respectively [6]. For backs, three of the top five most severe were hamstring muscle, MCL, and ACL injuries causing 1176, 870 and 815 days absence, respectively [6]. Knee injuries in particular (ACL and MCL) resulted in the greatest absence for forwards and backs [6]. At the 2007 RU World Cup, muscle/tendon (mainly muscle) and ligament/joint (non-bone) were the third and fourth most severe injuries, with backs having a higher severity of both [14] (not tested statistically). At the 2011 RU World Cup, ligament/joint (non-bone) and muscle/tendon (mainly tendon) were the third and fourth most severe injuries with backs again having a higher severity of both [7] (not tested statistically). Fuller et al. [8] identified knee ligament injuries as the most severe and Achilles tendon injuries as the fourth most severe at the 2015 RU World Cup for all players. In Williams et al.’s [11] meta-analysis, a similar pattern was seen, with ligament/joint (non-bone) injuries the second most severe and muscle/tendon injuries the fourth. Table 2 summarises the severity of muscle/tendon and ligament/joint (non-bone) injuries for English Premiership and World Cup competitions. The large variability can be attributed to several factors such as different settings (league or cup tournament), cohort sizes and opportunities for data collection.

For injury burden (days absence/1000 h), in the English Premiership competition across 2011–2018, ligament/joint (non-bone) injuries dominated the top five highest risk match injuries. Three different ligament injuries were usually in the top five highest risk injuries (all except 2015–2016 and 2017–2018 when there were two), with ACL and MCL injuries included every season (apart from 2017–2018 when ACL was not) [15,16,37,38]. Figure 2 features the top five highest risk injuries during 2011–2018.

In elite rugby, muscle/tendon and ligament/joint (non-bone) injuries are some of the most severe and frequently occurring injuries players receive and are, therefore, extremely debilitating to playing squads. Generally, in elite RU, there appears to be a trend towards more severe injuries [39]. Whether this is due to the more conservative approach to injury management or increased damage caused by larger collisions remains to be established. A deeper understanding of the potential causes and, subsequently, any preventative measures against these injuries would be of great value to both governing bodies and medical staff.

## 3. Risk Factors for Injury in Rugby

From the available literature, it is difficult to state exactly how each muscle/tendon and ligament/joint (non-bone) injury occurred during rugby matches or training. Nevertheless, the most common causes of injury in RL and RU are tackles and physical collisions [42], with the ball carrier generally at highest risk [43], though not for concussion [44]. Further risk factors for injury in rugby identified in previous literature are: playing position [5,6], level of play [11], training volume and load [45,46,47], ground conditions and playing surface [48], anthropometric characteristics [49,50], previous injury [51] including concussion [52,53], physiological characteristics [11,50] and age [54]. The precise mechanisms of tendon and ligament injury are not well understood [55,56], with multiple factors probably involved [55,56]. It has been suggested that interactions between genetic and environmental factors can amplify intrinsic risk factors (anthropometry, physiological characteristics, etc.) and place a predisposed athlete at higher risk of injury once an inciting event occurs [55,57,58]. For example, during typical physiological environments the matrix of ligaments and tendons will adapt in response to load [59]. However, variation in the loading pattern such as higher strains or a higher volume of low strains could lead to maladaptation, resulting in degeneration or a failed healing response [59], and thus injury (Figure 3). The tolerable load varies between individuals, resulting in large inter-individual variation in response to ligament and tendon tissue loading. This large inter-individual variation is thought to be partly due to a genetic component [60], meaning some individuals are more predisposed to ligament and tendon injury than others. 

## 4. Tendon and Ligament Pathologies

### 4.1. Tendinopathy

Tendons, especially the Achilles, are designed to tolerate significant loads. Mechanical loading of tendon leads to an increase in collagen gene expression and an upturn in collagen protein complex synthesis, which is likely regulated by the strain experienced by local tenocytes [62]. The increased collagen formation peaks ~24 h after substantial mechanical loading, while the degradation of collagen proteins also increases after loading but appears to peak earlier [62]. Thus, maintaining tendon homeostasis is a finely tuned process, and despite a tendon’s ability to adjust to mechanical loading, overuse will potentially result in injury such as tendinopathy. 

Traditionally, “tendinitis” was the preferred term to describe chronic pain in a symptomatic tendon, which implied that inflammatory processes played a central role in the disease aetiology. However, treatment protocols designed to modify inflammation had limited success [63,64] and few or no inflammatory cells were found in symptomatic tissue [65,66]. Therefore, the terms “tendinosis” or more generally “tendinopathy” are now preferred [67]. Tendinopathy is a diverse clinical syndrome associated with swelling, pain, impaired tissue healing and decreased performance [68]. There appears to be a continuum between physiology and pathology; as such, overuse (e.g., excessive repetitive loading of the tendon) could be considered the primary cause of disease [67].

Biomolecular studies of tendinopathy are relatively sparse although some observations have been made. Increased expression of messenger RNA (mRNA) has been found for type I and III collagens within symptomatic tendons [62,69]. This could reflect decreases in total collagen content (and a biological attempt to compensate) and an increased ratio of type III collagen relative to type I [70,71,72]. This increased proportion of type III collagen within the main fiber bundles appears to reduce fibril diameter [73], probably weakening the tendon and increasing risk of rupture [74]. 

Tendinopathies are caused by multiple intrinsic and/or extrinsic risk factors [55]. Common intrinsic risk factors include age, anthropometry, sex, anatomical factors, hyperthermia, previous injury and systemic diseases [75,76], with genetic variation also recently proposed [77]. Common extrinsic risk factors include environmental conditions, shoes/surface, training errors, nutrition, medication and mechanical loading [75,76,77]. Anatomical factors such as alignment and suboptimal biomechanics could contribute to two-thirds of Achilles tendon disorders among athletes [78]. Low-level highly repetitive strains below the failure threshold, or high strains even without great repetition, cause tendon degeneration [79]. Thus, excessive loading during physical training is considered the primary extrinsic determinant of tendon degeneration [80]. In the presence of intrinsic risk factors such as genetic predisposition, excessive loading may therefore further increase the risk of tendinopathy. There is no direct evidence of what causes tendinopathy in rugby players, although potential causes include: differing ground conditions that change the magnitude and temporal characteristics of the loads experienced; running and certain contact situations that elevate low-level repetitive loading; tackling, scrums, and mauls that elicit high strain; excessive training and match volume (insufficient recovery and/or excessive loading).

### 4.2. Tendon Rupture

Tendon rupture is an acute injury where partial or complete tearing of the tendon occurs. This is observed at the microscopic and macroscopic level, whereas tendinopathy occurs without macroscopic tearing [81]. Partial or complete rupture will inhibit tendon continuity, limiting range of motion and force-generating capabilities. Extrinsic risk factors are thought to dominate tendon rupture incidence, with intrinsic risk factors also considered important [82]. Intrinsic and extrinsic risk factors for tendon rupture are similar to those mentioned for tendinopathy, although rupture often follows one isolated overloading event [62,83,84,85,86,87]. In rugby, this is probably through high-loading scenarios such as scrums, mauls, sprinting, tackling and landing from jumps. During loading, the crimping formation of the collagen within the tendon is lost, and the collagen responds to the increasing load linearly [88]. Tendon strain >4% causes microscopic tearing of fibers and strain >8-10% causes macroscopic failure and rupture [88,89]. The aetiology of tendon rupture is not completely understood [90]. However, it appears to be multi-factorial, typically involving a combination of excessive loading and intrinsic risk factors [91]. Histologically, degenerative tendinopathy is the most frequent finding in acute tendon ruptures [92]. 

### 4.3. Molecular Changes in Tendinopathy and Tendon Rupture

Gene expression is altered in symptomatic tendons. Increased mRNA expression has been reported for proteoglycans such as aggrecan and biglycan [93], decorin and versican [94], glycoproteins such as tenascin-C and fibronectin [62], angiogenic factors such as vascular endothelial growth factor (VEGF) [95], collagen type I [94], tissue inhibitor of metallaoproteinase 1 (TIMP 1) and 2 [94], and proteolytic enzymes such as the disintegrin and metalloproteinase (ADAM-12) [96], plus several matrix metalloproteinases (MMPs 1, 2, 9, 13, and 23) [94,96]. Conversely, decreased mRNA expression has been reported for TIMP3 and MMPs 3, 10, and 12 [96]. However, the molecular signature of tendinopathy appears quite different from that of tendon rupture. Jones et al. [96] found lower mRNA expression in ruptured than tendinopathic tendons of ADAMTS 2, 3 and 17, MMP 7, 16, 23, 24 and 28, as well as TIMP 2, 3, and 4, and increased expression of ADAMs 8 and 12, A disintegrin and metalloproteinase with thrombospondin motifs 4, TIMP1, and MMPs 1, 8, 10, 12, 19, and 25. Such differences in gene expression potentially contribute to disease pathophysiology [97].

Alterations in gene expression in symptomatic tendons suggests there is an interaction between genes and environment and thus a genetic component to the aetiology of this disease. Indeed, in a twin study of tennis elbow (epicondylitis) in women [98], heritability was estimated at ~40%. Furthermore, several studies report associations between Achilles tendinopathy and several genetic variants, as discussed in Section 6. 

### 4.4. ACL Tear and Rupture

Injuries to the ACL are among the most frequent knee ligament injuries in sport and usually require reconstruction [99,100]. In the RU, although ACL injuries are not the most frequent, they have been in the top five most severe injuries for six of the last seven seasons in English Premiership Rugby (2011–2018) [37,39], accounting for 224 days of absence/1000 h in 2016–2017 [37,39]. Frequently, ACL injuries lead to muscle weakness, altered movement, joint effusion, reduced functional performance, and have been associated with continuing clinical sequelae such as chondral lesions, meniscal tears and increased risk of early-onset post-traumatic osteoarthritis [101,102,103,104,105]. 

Dallalana [106] established that the primary mechanisms of ACL injury in RU are a player being tackled, tackling or in general collisions, accounting for 43%, 29% and 14% of all ACL injuries, respectively. However, the remaining 14% of ACL injuries occurred through non-contact mechanisms such as twisting and turning [106]. More recently, using video to analyze the mechanisms for ACL injury in RU showed that 57% occurred through contact [107]. Two main scenarios were identified: offensive running and being tackled, suggesting that the ball carrier is at increased risk of ACL injury. The remaining 43% were through non-contact mechanisms, mainly sidestepping maneuvers. There are numerous intrinsic and extrinsic risk factors for ACL injury, including age, anthropometry, sex, previous injury, anatomical variation, neuromuscular and cognitive factors, and genetics [56,108,109,110] for intrinsic risk factors. Whereas for extrinsic risk factors, environmental conditions, shoes/surfaces, training errors, and mechanical errors would be common [56,108,109,110]. However, trauma to the knee is a fundamental requirement [108,109]. 

It is possible for ACL injuries to be either partial tears or complete ruptures. Like tendon, when load is placed through the ACL the crimping formation of collagen will stretch linearly with increasing load [111]. Strains >4% cause microscopic tearing of the fibers and >8–10% strain cause macroscopic failure and rupture [111]. Though high-traumatic strains are a typical cause of ACL rupture, microscopic damage to ligament tissue occurs at relatively low levels of strain [112]. Furthermore, changes at the microscopic level such as extra-cellular matrix (ECM) alterations and cellular damage alter the mechanical properties of ligaments, thus when a ligament with microstructural alterations has strain applied, rupture can follow [112]. Thus, ACL rupture may occur in the same manner as tendon rupture, with prior degeneration of the tissue before the inciting event. 

### 4.5. Molecular Characteristics of ACL Tear and Rupture

Over the last ~25 years, numerous studies have examined genetic factors that potentially predispose an individual to ACL injury [113,114,115,116,117,118,119,120,121,122,123,124,125,126,127,128,129]. Tears of the ACL seem at least twice as likely in individuals with a family history of ACL tear compared to those with no family history [113,114]. To the authors’ knowledge, there have been no twin studies estimating the heritability of ligament injury, unlike tendon [98]. In our opinion, this would be an extremely useful addition to the literature to develop an understanding in this area. The majority of research into the genetics of ACL injury has utilized gene association studies (GAS). From these studies, variants in several genes have been associated with altered risk of ACL injury, as detailed in Section 6.

## 5. Genetics of Tendon and Ligaments

Genetic variation may have a strong influence on tendon and ligament structure and function, which could alter an individual’s risk of injury. Inter-individual variability of tendon and ligament properties is likely to cause microtrauma and macrotrauma at differing strain levels among individuals, thus similar injury-inciting events amongst rugby players may have vastly different outcomes. Published associations exist between gene variants (of proteins that play structural and functional roles within tendons and ligaments) and susceptibility to injury for tendinopathy [130,131,132,133,134,135,136,137,138,139,140], tendon rupture [140] and ACL rupture [116,117,118,119,120,121,122,125,127,128,129,139]. Therefore, due to the high incidence and severity of tendon and ligament injuries within elite rugby, there is a potential future role for genetic screening of players to aid in injury risk management, but the practicalities are yet to be developed. In addition, the literature regarding genetic variants and tendon and ligament injuries is in its infancy, with little replication. Currently, there are no studies examining the genetics of tendon and ligament injuries within elite rugby. 

## 6. Identifying Candidate Genes

Traditionally, top-down or unmeasured genotype approaches have been utilized to identify the heritability of phenotypes. While these provide useful estimates for identifying the genetic influence of certain phenotypes, they offer no evidence of the specific genes or polygenic profiles that contribute to the phenotype. Furthermore, high-throughput approaches such as genome-wide association studies (GWAS) frequently identify a variety of candidate genes, of which only a small percentage are actually relevant to the phenotype of interest and validating all the identified candidate genes is not always possible [141]. Genome-wide association studies also require particularly large sample sizes to be effective and meet the generally accepted significance level of *p* < 5 × 10^−8^ to minimize the risk of false positives, but that is not yet feasible in rugby. Thus, there is a need to study candidate genes because an adequately powered GWAS is currently impossible, although judicious use of GWAS results from other relevant populations to identify candidate genes can be fruitful [142]. A strength of GAS is that selection of candidate genes is based on detailed knowledge of a protein and its role vis-à-vis the phenotype of interest. Once a candidate gene is identified, the next logical step is to find functionally significant polymorphisms, with priority given to non-synonymous (missense) single nucleotide polymorphisms (SNPs) that change an amino acid in a protein or a nonsense variation that creates a premature stop codon, as these are most likely to have substantial biological effects [143]. However, polymorphisms in regions of DNA that regulate the expression of genes have recently become more appreciated for their functional roles [144,145]. Thus, several genes have been identified that may influence injury risk and are worthy of study within elite rugby (Table 3).

### 6.1. COL1A1 as a Candidate Gene

The gene *COL1A1* codes for the α1 chain of Col I, which is responsible for the high tensile strength of tendons and ligaments via its strong parallel fiber bundles and cross-linking formation [146]. Several studies have investigated associations between the Sp1 polymorphism (rs1800012) and a variety of soft tissue injuries; including cruciate ligament ruptures, Achilles tendinopathy and rupture, shoulder dislocation and tennis elbow (Table 4). Individuals of TT genotype appear to be at lower risk of cruciate ligament injury, particularly the ACL [115,116,122]. In contrast, there seems to be no association between tendinopathies or tendon rupture and the Sp1 polymorphism [147]. 

### 6.2. COL3A1 as a Candidate Gene

The protein Col III is an important fibrillar collagen that is similar in structure to Col I. However, Col III is a homotrimeric molecule (three α1 (III) chains) as opposed to the heterotrimeric form of Col I [149,150]. Col III frequently mixes with Col I to form mixed fibrils and is also plentiful in elastic tissue [151]. Specifically, it is found in the solid component of tendons and ligaments [152], where it functions with Col I, V, and XII to enable normal collagen fibrillogenesis [153,154]. The pro-α1 chains of Col III are encoded by the *COL3A1* gene. Three studies have investigated the association between *COL3A1* and ACL rupture (Table 5), but none have examined tendon pathology. Stępień-Słodkowska et al. [129] found the AA genotype of the *COL3A1* rs1800255 polymorphism was more common in male recreational Polish skiers with ACL rupture than apparently healthy skiers. Similar evidence was found in Polish professional footballers [127], but not replicated in a broader population [155]. Collectively, these results suggest that individuals involved in sport who carry the AA genotype may have increased risk of ACL rupture. 

### 6.3. COL5A1 as a Candidate Gene

Probably the most explored gene regarding tendon and ligament injury is *COL5A1* (Table 6), which encodes the α1 chains of type V collagen. The protein Col V is a minor fibrillar collagen that is known to associate with type I and III collagen [156]. Although Col V is a minor collagen in terms of content, research suggests that it functions as a major collagen in developing connective tissues [157]. Mokone et al. [131] were the first to associate the *COL5A1* gene with Achilles tendon pathology, finding the C allele of the rs12722 polymorphism less common in those with injury. This association was replicated for Achilles tendinopathy [158] and ACL rupture in females [117], with the C allele also underrepresented in tennis elbow patients versus controls [159]. These findings suggest the C allele may be protective against tendon and ligament injuries. A recent investigation by the RugbyGene project [160] found differences in allele and genotype frequencies for the *COL5A1* rs12722 and rs3196378 polymorphisms between elite rugby athletes (rs12722: CC genotype = 21%, C allele = 47%; rs3196378: CC genotype 23%, C allele = 48%) and non-athletes (rs12722: CC genotype: 16%, C allele = 41%; rs3196378: CC genotype = 16%, C allele = 41%, *p* ≤ 0.02) [161]. These findings suggest that elite rugby players may have an inherited resistance against soft-tissue injury.

### 6.4. MIR608 as a Candidate Gene

MicroRNAs (miRNA) are a class of small non-coding RNAs that induce gene silencing and translational repression [164,165]. Allele-specific polymorphisms within miRNA target sites influence the tissue-specific miRNA regulation of hundreds of genes, which implies that their genetic variation may be a prevalent cause of inter-individual phenotypic variability [166]. This potential variance has been seen in the microRNA 608 (*MIR608*) gene, which was associated with altered risk of Achilles tendinopathy [135,155,163]. To date, three studies have investigated the link between the *MIR608* rs4919510 polymorphism and Achilles tendon pathology (Table 7), with none examining ACL rupture. The latest investigation involved a genome-wide approach; Kim et al. [155] observed that although *MIR608* rs4919510 did not approach genome-wide significance (*p* < 5 × 10^−8^), when covariates such as age, sex and ancestry were not used in analysis of a tentative association identified (*p* = 5.1 × 10^−3^). The combined results from the three studies suggest that *MIR608* may have a role in altering tendon injury risk but the evidence is inconclusive.

### 6.5. MMP3 as a Candidate Gene

The protein MMP3, encoded by the *MMP3* gene, has a fundamental role in the regular development, repair, and remodeling of connective tissues, by regulating ECM homeostasis via proteolytic activity [167]. Several studies have examined the association between polymorphisms rs679620, rs591058 and rs650108 within *MMP3* and Achilles tendon pathologies and ACL ruptures (Table 8). These three polymorphisms span most of the *MMP3* gene as they are within all four major haploblocks (one exon SNP rs679620, two intron SNPs rs591058, rs650108) [167]. Raleigh et al. [132] first investigated the three polymorphisms, finding all three independently associated with increased risk of Achilles tendinopathy, specifically the GG genotype of rs679620, CC genotype of rs591058, and AA genotype of 650108. The GG genotype of rs679620 has also been associated with Achilles tendon rupture [140]. Conversely, Posthumus et al. [120] and Gibbon et al. [168] found no independent associations between any of these variants and Achilles tendinopathy [168] or ACL rupture [120,168]. However, when inferred haplotype was considered, Posthumus et al. [120] and Gibbon et al. [168] found they were associated with ACL rupture and Achilles tendinopathy, respectively. Interestingly, Gibbon et al. [168] found the G (rs679620), C (rs5901058) and G (rs650108) alleles were overrepresented in controls, which contrasts with previous findings [132] but aligns with a recent study of Achilles tendon rupture, ACL tears and tendinopathy in a broader population [155]. Therefore, the literature appears to suggest the chromosomal region 11q22 has some influence on musculoskeletal injuries, most likely polygenic in nature, and warrants further investigation.

### 6.6. TIMP2 as a Candidate Gene

The TIMPs are natural inhibitors of MMPs, which they bind with in a 1:1 stoichiometry [169]. In pathological conditions such as Achilles tendinopathy where irregular MMP activity occurs, alterations in TIMP are important as they directly influence MMP activity [169]. The SNP *TIMP2* rs4789932 was associated with Achilles tendon pathologies in two studies [137,140] (Table 9). However, they contain opposing findings with the CT genotype associated with Achilles tendon pathology one [137], but overrepresented in controls in another [140]. Recently, Kim et al. [155] reported no association after corrections for testing multiple hypotheses, but possibly adds a little support to the data of El Khoury et al. [140]. Thus, although at present it is unclear which genotype/allele within the *TIMP2* polymorphism affects tendon injury risk, the evidence tentatively suggests that it may play a role.

### 6.7. VEGFA as a Candidate Gene

Angiogenesis is essential during the repair and remodeling of injured tendons, although it can also potentially reduce mechanical stability due to the proteolytic activity in the ECM by invading endothelial cells [170]. Vascular endothelial growth factor (VEGF) is an endothelial cell mitogen that stimulates angiogenesis [171,172]. It activates endothelial cells and vascular smooth muscle migration and proliferation, as well as enhancing endothelial cell survival and differentiation [173]. Vascular endothelial growth factor has a number of isoforms (A–D); the most relevant being VEGFA [173] encoded by the *VEGFA* gene. Variants within *VEGFA* have been associated with ACL rupture [125] and Achilles tendinopathy [174] (Table 10). Interestingly, a polymorphism appears to play a different role in acute (ACL rupture) and chronic (Achilles tendinopathy) injury. The CC variant of rs699947 was overrepresented in non-contact ACL ruptures compared to controls, suggesting a role in increased ACL rupture risk [125]. Yet the CC variant might protect against Achilles tendinopathy, being underrepresented in a control population versus an Achilles tendinopathy group [174]. Further investigation is needed to improve understanding of its role in musculoskeletal injury.

### 6.8. Additional Candidate Genes of Interest

Several genetic variants recently identified in a GWAS [155] are worthy of future study, such as *COLGALT1* rs8090 and *NID1* rs4660148. These had strongest associations with ACL rupture (*p* = 6 × 10^−4^) and Achilles tendon injury (*p* = 5 × 10^−5^), respectively, although none approached genome-wide significance (*p* < 5 × 10^−8^). Inevitably, there will be many other as yet unidentified genetic variants that emerge as research advances. 

## 7. Future Directions/Conclusions

The exact pathophysiology of tendon and ligament injuries is yet to be fully elucidated, as they are complex multifactorial conditions. There appears to be growing evidence of a genetic influence, although much stronger evidence is needed. The genes mentioned within this text and many others should be explored further regarding their relevance to tendon and ligament injuries. This would be particularly useful in a sport such as rugby, due to its high incidence and severity of injury. 

To be truly relevant to elite rugby, research must involve appropriate cohorts who possess the extreme phenotypes and behaviors only found at the elite level. Elite athletes undergo heavy training loads and are likely to exhibit characteristics near the limits of human physiological capability; indeed, elite rugby has one of the highest incidences of injury in sport, with tendon and ligament injuries some of the most frequent and severe. Regular participation at the elite level in rugby would mean players have been exposed to one of the highest levels of risk for tendon and ligament injury in any professional sporting environment, and at least to some extent, have been able to succeed in that sport despite that high environmental risk. This ability to recover from or withstand musculoskeletal soft tissue injury that is potentially performance-limiting or career-ending, but nevertheless achieve elite status, may be reflected in distinct genetic characteristics. Large sample sizes are required for genetic research to gain sufficient statistical power and reduce the likelihood of statistical errors. Additionally, the sample sizes should be in the hundreds and ideally thousands (especially if GWAS or other hypothesis-free approaches are to be used), which is extremely challenging due to the limited number of elite athletes in a given sport. Therefore, large international collaborations are required to achieve this aim within rugby [160]. Accordingly, genetic analyses of players already included in large rugby injury databases could prove fruitful in explaining some of the currently unexplained inter-individual variability in injury susceptibility and may provide new markers of injury risk within elite rugby. Such findings could then be applied alongside existing non-genetic data to aid the personalized management of playing load and injury risk amongst rugby players.

## Figures and Tables

**Figure 1 sports-07-00138-f001:**
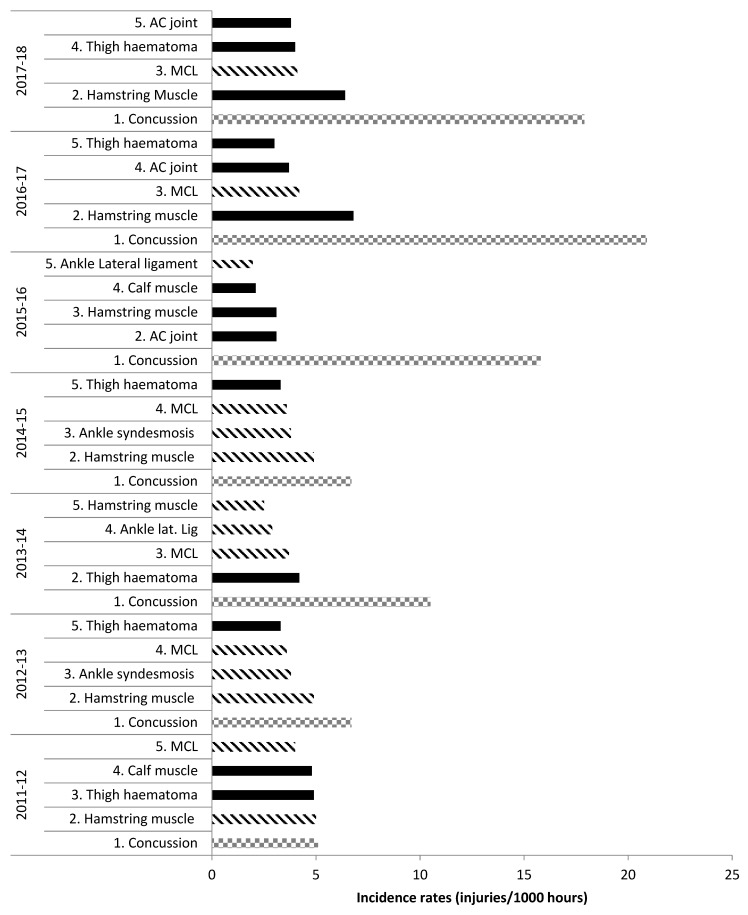
Top five most common injuries: English Premiership Rugby. Adapted from the Professional Rugby Injury Surveillance Project (PRISP) annual reports 2011–2018 [38,39]. Key: Lined bars = ligament injuries; squared bars = concussion; filled bars = any other injury.

**Figure 2 sports-07-00138-f002:**
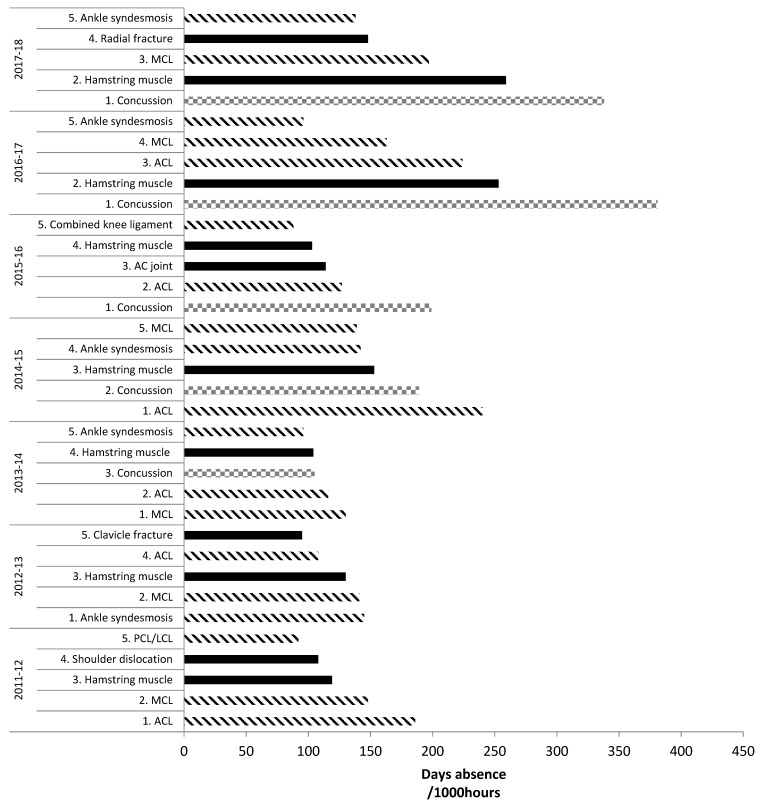
Top five highest risk match injuries: English Premiership Rugby. Adapted from PRISP annual reports, 2011–2018 [38,39]. Key: lined bars = ligament injuries; squared bars = concussion; filled bars = any other injury.

**Figure 3 sports-07-00138-f003:**
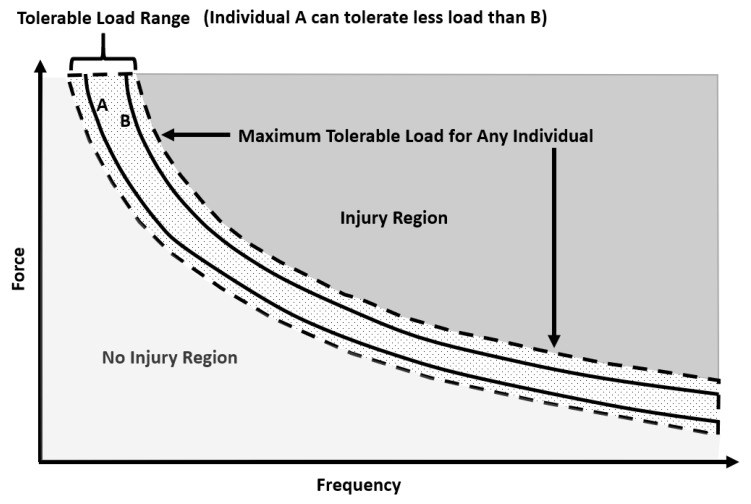
Hypothetical curve illustrating the relationship between magnitude (force) and frequency of load, which can injure the tendon or ligament (Injury Region). The tolerable load range for a given population is indicated by the dashed lines. The tolerable load curves for two hypothetical individuals are indicated by the solid lines (A can tolerate less load than B). Adapted from Reference [61].

**Table 1 sports-07-00138-t001:** Muscle/tendon and ligament/joint (non-bone) injury incidence rates in elite rugby union.

Study	Level	Injury Type		Match Injuries Incidence (Injuries/1000 Player Hours (95% CI))		
		Main Group	Sub-Group	Forwards	Backs	All
Brooks et al. [6] ^1,2^	English Premiership clubs ^4^	Muscle/Tendon	Strain/Tear/Rupture	14 *	20 *	17 *
		Ligament/Joint (non-bone)	Sprain/Rupture	13 *	15 *	14 *
Fuller et al. [14]	International	Muscle/Tendon	Muscle Rupture/Tear/Strain	18 (12–29)	27 (18–40)	22 (17–30)
			Tendon rupture/Tendinopathy	0	1 (0.2–8)	0.5 (0.1–4)
		Ligament/Joint (non-bone)	Sprain/Rupture	25 (17–37)	23 (15–36)	25 (18–33)
Fuller et al. [7]	International	Muscle/Tendon	Muscle rupture/Tear/Strain	20 (13–30)	20 (13–32)	20 (14–27)
			Tendon rupture/Tendinopathy	1 (0.1–7)	5 (2–12)	3 (1–6)
		Ligament/Joint (non-bone)	Sprain/rupture	22 (14–33)	18 (11–29)	20 (14–27)
Moore et al. [32]	International	Muscle/Tendon	Muscle strain/rupture	-	-	34 (23–49)
			Tendinopathy/rupture	-	-	9 (4–18)
		Ligament/Joint (non-bone)	Sprain/rupture	-	-	43 (31–61)
Fuller et al. [8]	International	Muscle/Tendon	-	34	39	-
		Ligament/Joint (non-bone)	-	25	34	
Williams et al. [11] ^3^	English Premiership clubs ^4^, Super 14 clubs ^4^, Vodacom cup 2008 clubs ^4^ and International	Muscle/Tendon	-	-	-	40 (21–76)
		Ligament/Joint (non-bone)	-	-	-	34 (18–65)

^1^ Study was before 2007 consensus statement on injury definitions and data collection procedures but used very similar methods. ^2^ No confidence intervals were annotated in the study. ^3^ Meta-analysis with seven studies used for pooled analysis [6,7,14,33,34,35,36]. ^4^ Top tier of professional rugby competitions in England, Australia, New Zealand, and South Africa. * These data were calculated by totaling all reported injuries within muscle/tendon or ligament/joint (non-bone), as Brooks et al. [6] only reported individual injuries rather than groups.

**Table 2 sports-07-00138-t002:** Muscle/tendon and ligament/joint (non-bone) injury severity rates in elite rugby union.

Study	Level	Injury Type		Match Injuries Severity (Days Absence (95% CI))		
		Main Group	Sub-Group	Forwards	Backs	All
Brooks et al. [6] ^1,2^	English Premiership clubs ^5^	Muscle/Tendon	Strain/Tear/Rupture	20 ^#^	16 ^#^	17 ^#^
		Ligament/Joint (non-bone)	Sprain/Rupture	22 ^#^	26 ^#^	24 ^#^
Fuller et al. [14]	International	Muscle/Tendon	Muscle Rupture/Tear/Strain	17 (10–25)	21 (9–33)	20 (12–27)
			Tendon Rupture/Tendinopathy	0	4 *	4 *
		Ligament/Joint (non-bone)	Sprain/Rupture	14 (8–20)	18 (9–27)	16 (11–21)
Fuller et al. [7]	International	Muscle/Tendon	Muscle Rupture/Tear/Strain	15 (8–23)	27 (16–38)	21 (14–28)
			Tendon Rupture/Tendinopathy	4 *	36 (0–92)	29 (0–75)
		Ligament/Joint (non-bone)	Sprain/Rupture	38 (8–68)	42 (12–72)	39 (18–61)
Fuller et al. [8] ^2,3^	International	Ligament/Joint (non-bone)	Knee Ligament	-	-	1507
			Achilles Tendon	-	-	188 *
Williams et al. [11] ^4^	English Premiership Clubs ^5^ and International	Muscle/Tendon	-	-	-	15 (5–24)
		Ligament/Joint (non-bone)	-	-	-	29 (19–39)

^1^ Study was before 2007 consensus statement on injury definitions and data collection procedures in the rugby union. ^2^ No confidence intervals reported in this study. ^3^ Study only reported injuries causing most days of absence rather than mean severity across a main injury group. ^4^ Meta-analysis with four studies used for pooled analysis [6,7,14,34]. ^5^ Top tier of professional rugby competitions in England. * Only one result in category. ^#^ These data are a “weighted severity” utilizing the following equation: incidence x severity/incidence for all individual muscle/tendon or ligament/joint (non-bone) injuries, as Brooks et al. [6] only reported individual injuries rather than groups.

**Table 3 sports-07-00138-t003:** Candidate genes, candidate proteins, and their abbreviations.

Candidate Protein	Candidate Protein Abbreviation	Candidate Gene	Candidate Gene Abbreviations
Type I collagen	Col I	*Collagen type I alpha I*	*COL1A1*
Type III collagen	Col III	*Collagen type III alpha I*	*COL3A1*
Type V collagen	Col V	*Collagen type V alpha I*	*COL5A1*
N/A Non-coding RNA	N/A	*MicroRNA 608*	*MIR608*
Matrix metalloproteinase-3	MMP3	*Matrix metalloproteinase-3*	*MMP3*
Tissue inhibitors of metalloproteinases-2	TIMP2	*Tissue inhibitors of metalloproteinases-2*	*TIMP2*
Vascular endothelial growth factor A	VEGFA	*Vascular endothelial growth factor A*	*VEGFA*
Nidogen 1	NID1	*Nidogen 1*	*NID1*
Collagen beta(1-O) galactosyltransferase 1	COLGALT1	*Collagen beta(1-O) galactosyltransferase 1*	*COLGALT1*

**Table 4 sports-07-00138-t004:** *COL1A1* rs1800012 genetic association studies with tendon and ligament injuries in humans.

Study	Phenotype	Target Population	Participants	Findings
Khoschnau et al. [115]	Cruciate ligament ruptures, shoulder dislocations	Sweden	No ethnicity reported. 233 cruciate ligament injury participants, 126 shoulder dislocation participants, 325 female controls	Individuals with TT genotype had a reduced risk of injury for cruciate ligament ruptures and shoulder dislocations compared to GG carriers.
Posthumus et al. [116]	ACL injuries	SA	Caucasian, 117 ACL rupture participants, 130 controls	TT genotype underrepresented in ACL injury group compared to controls.
Posthumus et al. [147]	Achilles tendinopathy, Achilles tendon ruptures	SA	Caucasian, 85 Achilles tendinopathy participants, 41 participants with partial or complete ruptures, 126 controls	No differences in genotypes.
Ficek et al. [122]	ACL injuries	Poland	Caucasian, 91 professional football players with ACL rupture—all non-contact, 143 apparently healthy professional soccer players as controls	No differences in genotypes. There was an overrepresentation of G–T haplotypes (1997G+1245T) in controls suggesting, carriers may have reduced risk of injury.
Stępien-Słodkowska et al. [121]	ACL injuries	Poland	No ethnicity reported, 138 male recreational skiers with ACL rupture, 183 apparently healthy male skiers as controls	Carriers of the GG genotype were at lower risk of ACL injury than carriers of the TT. genotype.
Erduran et al. [148]	Tennis elbow	Turkey	No ethnicity reported, 103 with tennis elbow, 103 apparently healthy controls	No differences in genotypes.

Key: SA = South Africa.

**Table 5 sports-07-00138-t005:** *COL3A1* rs1800255 genetic association studies with tendon and ligament injuries in humans.

Study	Phenotype	Target Population	Participants	Findings
Stephien-Slodkowska et al. [129]	ACL rupture	Poland	No ethnicity reported, 138 male recreational skiers with ACL ruptures, 183 male apparently healthy skiers	The AA genotype was overrepresented in the ACL group compared to controls.
O’Connell et al. [127]	ACL rupture	SA/ Poland	Caucasian. 333 participants with ACL rupture (242 SA and 91 Poland), 378 apparently healthy controls (235 SA and 143 Poland).	No differences in genotype frequency distributions between the SA ACL group and the SA control group. However, the AA genotype was overrepresented in the Polish ACL group compared to Polish controls. No allele associations for any of the groups.
Kim et al. [155]	ACL rupture		Caucasian, Latin-American, East Asian, African, South-East Asian. 5148 Achilles tendon injury participants, 97,831 apparently healthy controls, 598 ACL rupture participants, 98,744 apparently healthy controls,	No associations after Benjamini–Hochberg correction for testing multiple hypotheses.

Key: SA = South Africa.

**Table 6 sports-07-00138-t006:** *COL5A1* rs12722 genetic association studies with tendon and ligament injuries in humans.

Study	Phenotype	Target Population	Participants	Findings
Mokone et al. [131]	Achilles tendon pathology, Achilles tendinopathy, Achilles tendon rupture	SA	Caucasian, 111 participants with current or past history of Achilles tendon pathology, including 72 chronic tendinopathy participants, 39 Achilles tendon rupture participants	The frequency of the A2 (C) allele was higher in the controls compared to the Achilles tendon pathology group. An even stronger protective role was seen for the A2 (C) allele in in controls compared to the chronic tendinopathy patients.
September et al. [158]	Achilles tendinopathy	SA/Australia	Caucasian, 83 Australian and 93 SA tendinopathy patients, 210 Australian, and 132 SA controls	Individuals with CC genotype in both populations (Australian/SA) had a reduced risk of developing Achilles tendinopathy compared to any other genotypes.
Posthumus et al. [162]	ACL injuries	SA	Caucasian, 129 ACL rupture participants, 216 physically active controls with no history of ACL injury	The CC genotype was underrepresented in the female ACL rupture group, but not in the male.
Stepien–Slodkowska et al. [128]	ACL injuries	Poland	No ethnicity reported, 138 male recreational skiers with ACL ruptures, 183 apparently healthy male recreational skiers without any reported history of ligament or tendon injury.	No differences in genotype distribution between groups. Higher frequency of rs12722 C-T and rs13946 C-T polymorphisms haplotype in controls suggests reduced risk of ACL injury.
Altinisik et al. [159]	Tennis elbow	Turkey	No ethnicity reported, 152 tennis elbow patients, 195 healthy controls.	Individuals with the A2 (C) allele were underrepresented in patient group. Individuals with A1 allele (T) have an increased risk of developing tennis elbow.
Brown et al. [163]	Achilles tendinopathy, Achilles tendon rupture	UK	Caucasian, 87 Achilles tendinopathy participants, 25 Achilles tendon rupture participants, 130 asymptomatic controls	No independent differences found between groups. Three inferred allele combinations from rs12722, rs3196378, and rs71746744 were identified as risk modifiers. The T–C–D combination was associated with increased risk of Achilles tendon pathology and rupture, the C–A–I combination was associated with increased risk of Achilles tendon pathology, tendinopathy and rupture, the C–C–D combination was associated with decreased risk of Achilles tendon pathology and rupture.
Kim et al. [155]	Achilles tendinopathy, Achilles tendon rupture		Caucasian, Latin-American, East Asian, African, South-East Asian, 5148 Achilles tendon injury participants, 97,831 apparently healthy controls, 598 ACL rupture participants, 98,744 apparently healthy controls	No associations after Benjamini–Hochberg correction for testing multiple hypotheses.

Key: SA = South Africa.

**Table 7 sports-07-00138-t007:** *MIR608* rs4919510 genetic association studies with tendon and ligament injuries in humans.

Study	Phenotype	Target Population	Participants	Findings
Abrahams et al. [135]	Achilles tendinopathy	SA/Australia	Caucasian, 160 chronic Achilles tendinopathy participants, 342 apparently healthy controls	The CC genotype frequency of rs4919510 was overrepresented compared to the CG and GG genotypes. The combined rs4919510 CC genotype and *COL5A1* rs3196378 CA genotype was overrepresented in the tendon group compared to controls. Furthermore, the rs4919510 CC genotype and the *COL5A1* rs3196378 A allele was overrepresented in the tendon group compared to controls.
Brown et al. [163]	Achilles tendinopathy and Achilles tendon rupture	UK	Caucasian, 112 Achilles tendon pathology participants (87 chronic Achilles tendinopathy and 25 Achilles tendon rupture, 130 apparently healthy controls	No differences in genotype frequency or allele frequency distributions between Achilles tendinopathy and controls. However, the CG genotype of rs4919510 was associated with decreased risk of rupture compared to controls. When inferred allele combinations were analyzed for rs4919510 and *COL5A1* rs3196378, and no associations found with risk of Achilles tendinopathy.
Kim et al. [155]	Achilles tendinopathy, Achilles tendon rupture		Caucasian, Latin-American, East Asian, African, South-East Asian, 5148 Achilles tendon injury participants, 97,831 apparently healthy controls, 598 ACL rupture participants, 98,744 apparently healthy controls	Moderate–weak evidence of replication (*p* = 5.1 × 10^−3^) for Achilles tendinopathy or rupture, but no replication with ACL rupture, after Benjamini–Hochberg correction for testing multiple hypotheses.

Key: SA = South Africa.

**Table 8 sports-07-00138-t008:** *MMP3* rs679620, rs591058 and rs650108 genetic association studies with tendon and ligament injuries in humans.

Study	Phenotype	Target Population	Participants	Findings
Raleigh et al. [132]	Achilles tendinopathy and rupture	SA	Caucasian, 114 Achilles tendon pathology patients including 75 with Achilles tendinopathy and 39 with partial or complete rupture, 98 controls	Independent associations between the GG genotype of rs679620, the CC genotype of rs591058, and the AA genotype of rs650108 and Achilles tendinopathy. The ATG haplotype (rs679620, rs591058, and rs650108) was under-represented in the tendinopathy compared to controls. No associations between *MMP3* variants and Achilles tendon rupture.
Posthumus et al. [120]	ACL rupture	SA	Caucasian, 129 ACL rupture patients, 216 apparently healthy controls	No independent associations for rs679620 compared to controls. Haplotypes T-1G-A-A and C-2G-G-G (MMP10 rs485055, MMP1 rs1799750, MMP3 rs679620, and MMP12 rs2276109) were different between control and ACL groups and controls and non-contact subgroup, respectively.
El Khoury et al. [140]	Achilles tendinopathy and rupture	UK	Caucasian. 118 Achilles tendon pathology patients including 93 with Achilles tendinopathy and 25 participants with partial or complete rupture, 131 asymptomatic controls	rs679620 GG genotype overrepresented in Achilles tendon rupture group compared to controls. No association with Achilles tendinopathy.
Gibbon et al. [168]	Achilles tendinopathy ACL rupture	Australia/SA	White Caucasian, 160 Achilles tendinopathy patients, 195 apparently healthy controls, 234 ACL rupture patients, 232 apparently healthy controls	No independent differences for rs679620, rs591058 and 650108 between Achilles tendinopathy and controls or between ACL rupture and controls. Haplotype 6a-G-C-G (rs3205058, rs679620, rs591058 and rs650108) overrepresented in the control group compared to the Achilles tendinopathy group when only Australian samples analyzed. No genotype or allele frequency differences from inferred haplotypes for ACL injury.
Kim et al. [155]	Achilles tendinopathy, Achilles tendon rupture		Caucasian, Latin-American, East Asian, African, South-East Asian. 5148 Achilles tendon injury participants. 97,831 apparently healthy controls, 598 ACL rupture participants, 98,744 apparently healthy controls	No associations after Benjamini–Hochberg correction for testing multiple hypotheses.

Key: SA = South Africa.

**Table 9 sports-07-00138-t009:** *TIMP2* rs4789932 genetic association studies with tendon and ligament injuries in humans.

Study	Phenotype	Target Population	Participants	Findings
El Khoury et al. [137]	Achilles tendinopathy and rupture	SA/Australia	Caucasian, 173 Achilles tendon pathology participants of which 134 with Achilles tendinopathy and 39 with partial or complete rupture, 248 asymptomatic controls	Association between *TIMP2* rs4789932 and Achilles tendinopathy. The CC variant was overrepresented within controls, while the CT variant was overrepresented within the combined Achilles tendon pathology group. No differences between the rupture group and controls.
El Khoury et al. [140]	Achilles tendinopathy and rupture	UK	Caucasian, 118 Achilles tendon pathology participants of which 93 had chronic Achilles tendinopathy and 25 participants with Achilles tendon rupture, 131 asymptomatic controls	Difference in genotype frequency between male Achilles tendon pathology compared to male controls. Further, difference between male ruptures compared to controls. The CT genotype was associated with lower risk of Achilles tendon pathology.
Kim et al. [155]	Achilles tendinopathy, Achilles tendon rupture		Caucasian, Latin-American, East Asian, African, South-East Asian 5148 Achilles tendon injury participants, 97,831 apparently healthy controls 598 ACL rupture participants, 98,744 apparently healthy controls.	No associations after Benjamini–Hochberg correction for testing multiple hypotheses.

Key: SA = South Africa.

**Table 10 sports-07-00138-t010:** *VEGFA* rs699947 genetic association studies with tendon and ligament injuries in humans.

Study	Phenotype	Target Population	Participants	Findings
Rahim et al. [125]	ACL rupture	SA	Caucasian. 227 ACL rupture participants, 227 apparently healthy controls with no history of ACL injury	The CC genotype of rs699947 was overrepresented in participants with non-contact ACL ruptures compared to controls. The rs1570360 GA genotype was overrepresented within controls. The A-A-G haplotype (rs699947, rs1570360 and 2010963) was overrepresented in the control group compared to the non-contact ACL group.
Rahim et al. [174]	Achilles tendinopathy	SA/UK	Caucasian, 195 chronic Achilles tendinopathy participants (87 from SA, 108 UK), 250 asymptomatic controls (120 SA, 130 from UK)	The CC genotype of rs699947 was overrepresented in the SA control group compared to the SA tendinopathy group. No other independent frequency differences found. The *VEGFA* A-G-G inferred haplotype (rs699947, rs1570360, and rs2010963) was associated with increased risk of tendinopathy in the SA group and the SA and UK combined group.
Rahim et al. [175]	ACL rupture	SA	SA colored ethnic group (unique to Western Cape of SA), 98 ACL rupture participants, 100 physically active asymptomatic controls with no history of tendon or ligament injury	No differences in genotype or allele frequency data for any of the VEGFA polymorphisms studied. Further, no associations found from inferred haplotype analysis.

Key: SA = South Africa.

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
