# Peer review of "Tendon and Ligament Injuries in Elite Rugby: The Potential Genetic Influence"

_sports, 2019, doi:10.3390/sports7060138_

Round 1
Reviewer 1 Report
The study reviews tendon and ligament injury incidence and severity within elite rugby union and rugby league, and also discusses the biological make up of tendons and ligaments and how genetic variation may influence this and predisposition to injury.
The results of this research are in my opinion of relevance to the field of sports science and would fit the scope of the journal. There are, however, minor concerns which should be addressed.
Specify the type of review used.
Page 3, line 78: “The aim of this review therefore is …”
Please, include the Method section in this paper and provide information on a literature searching strategy.
Author Response
Thank you for taking the time to review our work and we very much appreciate your feedback. Below are responses to your suggestions:
Specify the type of review used.
Page 3, line 78: “The aim of this review therefore is …”
Response:
Thank you for your comment, we have placed the word ‘narrative’ within this sentence.
Please, include the Method section in this paper and provide information on a literature searching strategy.
Response:
As this is a narrative review, we do not feel a specific method section is appropriate for this article. However, we have included a guide as to how we performed our literature search from line 85-90:
“A structured literature search was performed for empirical research studies and review articles. The search terms included “elite rugby injury”, “injury mechanisms in rugby”, “tendon and ligament epidemiology”, “pathophysiology of tendon and ligament injury”, “molecular characteristics of tendon and ligament injury”, “genetics of tendon and ligament injury”. The reference list of all articles were also examined for eligible studies to ensure no relevant articles were omitted from the search.”
Reviewer 2 Report
SPORTS 513177
Tendon and ligaments injuries in elite rugby: the potential genetic influence
This study reviewed tendon and ligament injury incidence and severity within elite rugby union and rugby league players. In addition, the study presents biological make up of tendons and ligaments and discusses how genetic variation may influence predisposition to injury.
The main concept of the study is very interesting, and the authors are amended for this extensive literature review. However, there are some issues that should be addressed:
General comments
a) although this study is not a systematic review, I think that the authors should mention some criteria used to identify the examined studies (e.g keywords for the main parts of your review, dates of publication et.c).
b) Please, consider diminishing the number of Tables/Figures in your manuscript. The authors refer to findings that are relevant enough to be mentioned but not all of them should be depicted in a figure or a table.
Abstract
1. The abstract of your manuscript should be a guide to the most important parts of your manuscript’s content. Many readers will only read the Abstract of this manuscript. I feel that the Abstract is written more like an Introduction and is not summarizing what the authors found and why these findings are useful and important. Please, correct.
2. Keywords should represent the content of your manuscript and be specific to your field but should not appear in the title of your manuscript. Please, correct.
Introduction
3. Lines 76-78: please provide a clearer rationale for this study. Why this study is important?
Tendon and Ligament Injury Incidence Rates and Severity in Rugby
4. Line 85: I think that the verb is missing from this sentence.
5. I think that Figure 1 and Figure 2 provide almost the same information for the reader: the most common and the top five highest risk injuries in English Premiership Rugby. One of the Figures could be deleted from the study without losing important information.
Risk Factors for Injury in Rugby
6. Lines 198-199: I think these lines belong to the ‘injury incidence rate and severity’ part of the manuscript.
Tendon and Ligament Pathologies
7. Please consider removing Table 3 and Table 4, from the manuscript. Relevant information is already provided and can be added in the Text.
Identifying Candidate Genes
8. Please, consider removing Table 5, as relevant information is provided in Tables 6-12.
9. Please, consider reducing information presented in lines 473-483, as it is also presented in Table 9.
References
10. Please, correct the references according to the journal’s instructions
Author Response
Thank you for taking the time to review our work and we very much appreciate your feedback. Below are responses to your suggestions:
General comments
a) although this study is not a systematic review, I think that the authors should mention some criteria used to identify the examined studies (e.g keywords for the main parts of your review, dates of publication et.c).
Response:
We have included a guide as to how we performed our literature search from line 85-90:
“A structured literature search was performed for empirical research studies and review articles. The search terms included “elite rugby injury”, “injury mechanisms in rugby”, “tendon and ligament epidemiology”, “pathophysiology of tendon and ligament injury”, “molecular characteristics of tendon and ligament injury”, “genetics of tendon and ligament injury”. The reference list of all articles were also examined for eligible studies to ensure no relevant articles were omitted from the search.”
b) Please, consider diminishing the number of Tables/Figures in your manuscript. The authors refer to findings that are relevant enough to be mentioned but not all of them should be depicted in a figure or a table.
Response:
We have removed two tables from the text.
Abstract
1. The abstract of your manuscript should be a guide to the most important parts of your manuscript’s content. Many readers will only read the Abstract of this manuscript. I feel that the Abstract is written more like an Introduction and is not summarizing what the authors found and why these findings are useful and important. Please, correct.
Response:
Abstract has been edited to include the candidate genes of interest identified by the review.
2. Keywords should represent the content of your manuscript and be specific to your field but should not appear in the title of your manuscript. Please, correct.
Response: Keywords have be altered.
Introduction
3. Lines 76-78: please provide a clearer rationale for this study. Why this study is important?
Response:
We have added a sentence to try and highlight the importance.
Tendon and Ligament Injury Incidence Rates and Severity in Rugby
4. Line 85: I think that the verb is missing from this sentence.
Response:
This sentence has been deleted based on another reviewer’s comments.
5. I think that Figure 1 and Figure 2 provide almost the same information for the reader: the most common and the top five highest risk injuries in English Premiership Rugby. One of the Figures could be deleted from the study without losing important information.
Response:
We politely disagree, figure 1 includes incidence rates (injuries/1000 hours), while figure 2 includes risk (days absence/1000 hours), with the make up of these figures containing different injuries.
Risk Factors for Injury in Rugby
6. Lines 198-199: I think these lines belong to the ‘injury incidence rate and severity’ part of the manuscript.
Response:
We feel this sentence fits well where it is currently placed. However, happy to remove if strongly opposed.
Tendon and Ligament Pathologies
7. Please consider removing Table 3 and Table 4, from the manuscript. Relevant information is already provided and can be added in the Text.
Response: Both tables have been removed and a small amount of text has been added to the main text.
Identifying Candidate Genes
8. Please, consider removing Table 5, as relevant information is provided in Tables 6-12.
Response:
We feel table 5 is important as it highlights the relationship between the protein and the gene with relevant abbreviations which is not addressed within the text or other tables.
9. Please, consider reducing information presented in lines 473-483, as it is also presented in Table 9.
Response:
Duplication of text has been removed and the paragraph shortened.
References
10. Please, correct the references according to the journal’s instructions
Response: Edits have been made to all references.
Reviewer 3 Report
General Comments:
In general, this is a clear, concise, and well-written manuscript. The introduction is relevant and theory based. Sufficient information about the previous studies findings is presented for readers to follow the present study rationale. Additionally, the authors make a systematic contribution to the research literature in this area of investigation. Overall, this is a high-quality manuscript that has implications for the theoretical basis and the development vision of rugby union and league, especially in the field of rehabilitation.
Minor corrections:
Line 86: There are two percentage injury rates mentioned for ligament/joint (25.2%) and muscle tendon (24.2%) and cited by Gissane (2003). Please, review the percentage again because I am not found these reported rates by Gissane (2003) study.
Line 119: In table (1), please review all values of Brooks (2005) regarding the injury incidence of rugby positions, and please refer in this table to the kind of rugby sport whether union or league.
Line 169: In table (2), please review all values of Brooks (2005) regarding the injury severity.
Line 313: Please delete the citation of reference number (106) because, at the beginning of Line 311, you have already cited.
Line 580: begin a new sentence with Additionally.
Author Response
Thank you for taking the time to review our work, we very much appreciate your comments. Below are responses to your suggestions:
Line 86: There are two percentage injury rates mentioned for ligament/joint (25.2%) and muscle tendon (24.2%) and cited by Gissane (2003). Please, review the percentage again because I am not found these reported rates by Gissane (2003) study.
Response:
Thankyou for spotting this, we have decided to remove this sentence in its entirety.
Line 119: In table (1), please review all values of Brooks (2005) regarding the injury incidence of rugby positions, and please refer in this table to the kind of rugby sport whether union or league.
Response:
The table title now includes the word ‘union’. Additionally, we have added a note to the table to identify how the data from the Brooks et al. study were calculated:
‘‘These data were calculated by totaling all reported injuries within Muscle/Tendon or Ligament/Joint (non-bone), as Brooks et al. [6] only reported individual injuries rather than groups.’
Line 169: In table (2), please review all values of Brooks (2005) regarding the injury severity.
Response:
We have added a note to the table to identify how the data from the Brooks et al. study were calculated:
‘These data are a ‘weighted severity’ utilizing the following equation: incidence x severity/incidence for all individual Muscle/Tendon or Ligament/Joint (non–-bone) injuries, as Brooks et al. [6] only reported individual injuries rather than groups.’
Thankyou for pointing this out.
Line 313: Please delete the citation of reference number (106) because, at the beginning of Line 311, you have already cited.
Response:
Deleted – thank you.
Line 580: begin a new sentence with Additionally.
Response:
Additionally added.
Reviewer 4 Report
The manuscript provides a deep revision of major interest for rugby. It is well written and structures, thus, suitable to be published in Sports.
I just have 2 main thought I would like to address:
1) why did the authors decide to perform a narrative review, instead of a systematic review?
2) I would suggest to rotate the figures 1 and 2, and make them vertical. It would be easier to understand if the injury was horizontal.
Small comment: L61 – please revise. Starting the sentence with unfortunately seems awkward, as the previous sentence is pejorative.
Author Response
Thank you for taking the time to review our work, we very much appreciate your comments. Below are responses to your points:
1) why did the authors decide to perform a narrative review, instead of a systematic review?
Response:
We decided to perform a narrative review rather than a systematic review, as this article includes three fairly distinct sections – Tendon and ligament injury within rugby, tendon and ligament pathologies and genetics of tendon and ligament. Therefore, we felt this article would be better suited and structured for a narrative review.
2) I would suggest to rotate the figures 1 and 2, and make them vertical. It would be easier to understand if the injury was horizontal.
Response:
Thankyou for your suggestion, we have rotated both figures and we feel they are easier to read based on your suggestion.
Small comment: L61 – please revise. Starting the sentence with unfortunately seems awkward, as the previous sentence is pejorative.
Response:
Word deleted